# Effects of Exogenous Regulation of PPARγ on Ovine Oocyte Maturation and Embryonic Development In Vitro

**DOI:** 10.3390/vetsci11090397

**Published:** 2024-08-28

**Authors:** Hengbin Yu, Yue Zhang, Yidan Zhang, Shuaitong Chen, Zhenghang Li, Wenhui Pi, Weibin Zeng, Guangdong Hu

**Affiliations:** College of Animal Science and Technology, Shihezi University, Shihezi 832003, China; 19915231430@163.com (H.Y.); zhangyue1@stu.shzu.edu.cn (Y.Z.); 13230661896@163.com (Y.Z.); 18290740107@163.com (S.C.); lizhenghang@stu.shzu.edu.cn (Z.L.); wzjpwh@163.com (W.P.)

**Keywords:** sheep, PPARγ, oocyte, lipid metabolism, oxidative stress

## Abstract

**Simple Summary:**

Lipid metabolism plays an important role in the development of oocytes and early embryos. Peroxisome proliferators-activated receptor γ is one of the key nuclear transcription factors that regulate lipid metabolism. Their main function is to regulate the expression of downstream genes and participate in the deposition and decomposition of intracellular lipids, thus regulating the level of intracellular lipid metabolism. Studies have shown that PPARγ can also be used as an antioxidant to reduce cell oxidative stress. Energy metabolism and oxidative stress are important factors affecting the development of oocytes and early embryos. In this study, the PPARγ agonist rosiglitazone was used to study the effects of PPARγ on oxidative stress and lipid metabolism in sheep oocytes and early embryos. The results showed that PPARγ could alleviate oxidative stress, improve the ability of oocytes to develop in vitro, and then improve the reproductive performance of sheep.

**Abstract:**

Lactating oocytes consume a lot of energy during maturation, a large part of which comes from lipid metabolism. PPARγ is a key regulator of lipid metabolism. In this study, rosiglitazone (RSG), an activator of PPARγ, was added to a mature medium to investigate its effects on the levels of spindle and the chromosome arrangement, lipid deposition, reactive oxygen species (ROS), and glutathione (GSH) levels, oocyte secretion factors, apoptosis and lipid metabolism-related gene expression, and subsequent embryonic development during the maturation of sheep oocytes. The oocyte secretion factor affects gene expression related to apoptosis and lipid metabolism and subsequent embryonic development. The results showed that the proportion of spindle and normal chromosome arrangements increased in the 5 μM RSG treatment group, the lipid content increased after cell maturation, the ROS level decreased, and the GSH level increased. The expressions of oocyte secretion factor (*GDF9* and *BMP15*), anti-apoptosis gene (*BCL2*), and lipid metabolism-related genes (*ACAA1*, *CPT1A*, *PLIN2*) were increased in the 5 μM treatment group. Finally, the development of blastocysts was examined. After the oocytes were treated with 5 μM RSG, the blastocyst rate and the gene expression of the totipotency gene (*OCT4*) were increased. It was concluded that increasing PPARγ activity during ovine oocyte maturation could promote lipid metabolism, reduce oxidative stress, and improve the ovine oocyte maturation rate and subsequent embryo development.

## 1. Introduction

Oocyte quality is essential for the successful reproduction of mammals and for early embryo development after in vitro fertilization. Mammalian oocyte maturation is a dynamic and complex process, which mainly starts with germinal vesicle breakdown (GVBD) and moves to nuclear maturation and first polar body (PBI) extrusion, and finally makes oocytes mature to the MII stage and wait for fertilization. The full development of oocytes requires simultaneous nuclear and cytoplasmic maturation, which is accompanied by a large amount of energy consumption. In oocytes, glycolysis is very weak and requires the metabolism of carbohydrates, lipids, and amino acids to produce sufficient energy to achieve full maturity [1]. Prior to oocyte maturation and early embryo implantation, the endogenous storage of glycogen in oocytes and early embryos is insufficient to maintain the energy expenditure required for maturation and early embryo development, and the protein is not used for ATP production [2]. In addition, lipid droplets accumulate continuously during the maturation of oocytes, and large amounts of ATP can be produced through fatty acid β-oxidation [3]. Therefore, it is speculated that lipids play an important role during this period. At the same time, lipid metabolism also plays a significant role in promoting oocyte maturation and improving oocyte quality [4,5].

As the largest cells in mammals, oocytes and early embryos often consume a lot of energy during their growth and development. As the main energy supplier, lipids accumulate in oocytes in the form of triglycerides to form special organelles—lipid droplets. In oocytes, lipid droplets combine with mitochondria to form energy-supplying units for mitochondrial β-oxidation [6]. More and more studies have shown that fatty acid oxidation is essential for the development of oocytes. In mouse oocytes, the lipid metabolism level was increased, the rate of fatty acid β-oxidation was up-regulated, and the maturation and early embryonic development of mouse oocytes were improved [4]. The inhibition of fatty acid β-oxidation during the in vitro maturation of bovine oocytes resulted in impaired oocyte development and reduced the ability of blastocyst development after in vitro fertilization [2]. In addition, fatty acids were involved in the composition of the phospholipid bilayer of the cell membrane, and in the assembly of the connection system between granulosa cells and oocytes during folliculogenesis [7].

Peroxisome proliferator-activated receptors (PPARs) belong to the nuclear receptors of the steroid receptor superfamily and are mainly involved in metabolism and inflammation in vivo, in addition to cell growth, differentiation, and apoptosis [8]. The PPAR family consists of three subtypes: PPARα, PPARβ/δ, and PPARγ. PPARγ is directly involved in the target genes that regulate lipid uptake, lipid synthesis, lipid storage, and lipolysis, thereby regulating the intracellular lipid metabolism process [9]. Previous studies have shown that the regulation of PPARγ expression regulates lipid metabolism in porcine oocytes, thereby improving the maturation of oocytes and the development of early embryos [10]. Knocking out PPARγ in mouse ovaries causes abnormal follicle rupture in mice, leading to impaired ovulation [11]. Studies on PPARγ in ovine oocytes are rarely reported, so the regulation mechanism of PPARγ in ovine oocytes is further elucidated by using its agonist, rosiglitazone.

## 2. Materials and Methods

### 2.1. Oocyte Collection and In Vitro Culture

Sheep ovaries collected from a local slaughterhouse were loaded into 0.9% normal saline at 37 °C and transported to the laboratory within 2 h. The follicles on the ovaries were cut with a surgical blade in the egg collection fluid, so that the cumulus oocyte complexes (COCs) were extruded and deposited in the Oocyte Collection Media at 37 °C. The COCs were washed 3 times in HEPES tissue medium (H-TCM199, Sigma-Aldrich, St. Louis, MO, USA) containing 1% fetal bovine serum (FBS, Thermo Fisher Scientific, Waltham, MA, USA). Each group of 50–80 COCs was cultured in a four-well plate (Thermo Fisher Scientific, Waltham, MA, USA) with 500 μL TCM199 (Earle’s equilibrium salt, 10% FBS, 1 mM SodiumPyruvate (Thermo Fisher Scientific, Waltham, MA, USA), 1× (2 mM) Glutamax (Thermo Fisher Scientific, Waltham, MA, USA), 5 μg/mL Folltropin (Solarbio, Beijing, China), 20 ng/mL Estradiol (Sigma-Aldrich, St. Louis, MO, USA), and 200 μg/mL Gentamicin (Solarbio, Beijing, China). The surface covered with paraffin oil (Thermo Fisher Scientific, Waltham, MA, USA), and cultured at 38.5 °C in 5% CO_2_ saturated humidity for 24 h. The mature media containing 0, 5, 10, and 20 μM RSG (MedChemExpress, South Brunswick, NJ, USA) were cultured with 30 COCs each for 24 h. Each experiment was repeated three times.

### 2.2. Parthenogenesis Activation and Embryo Culture

After the oocytes had matured (by observing the expulsion of polar body) for 24 h in vitro, they were gently blown through 1 mg/mL Hyaluronidase (Sigma-Aldrich, St. Louis, MO, USA) to remove cumulus cells. Thirty oocytes were transferred to HEPES-SOF (NaCl (Solarbio, Beijing, China) 107.7 mM, NaHCO_3_ (Solarbio, Beijing, China) 25.07 mM, CaCl_2_•2H_2_O (Solarbio, Beijing, China) 1.17 mM, KH_2_PO_4_ (Solarbio, Beijing, China) 1.19 mM, KCl (Solarbio, Beijing, China) 7.16 mM, MgCl_2_•6H_2_O (Solarbio, Beijing, China) 0.49 mM, GlutaMax (Thermo Fisher Scientific, Waltham, MA USA) 1 mM, Sodium Pyruvate (Thermo Fisher Scientific, Waltham, MA, USA) 0.4 mM, Myo-Inositol (Thermo Fisher Scientific, Waltham, MA, USA) 2.77 mM, MEM-NEAA (Sigma-Aldrich, St. Louis, MO, USA) 1%, BME-EAA (Sigma-Aldrich, St. Louis, MO, USA) 1%, Gentamicin (Solarbio, Beijing, China) 25 μg/mL, BSA (Solarbio, Beijing, China) 8 mg/mL) solution and cleaned 3 times, then transferred to HEPES-SOF solution with 5% ethanol and activated for 5 min. After activation, the oocytes were cleaned with HEPES-SOF solution 3 times and cultured in 2 mM 6-DMAP (Sigma-Aldrich, St. Louis, MO, USA) medium. After 4 h, the oocytes were cleaned with HEPES-SOF solution 3 times. They were placed in embryo culture medium and cultured at 38.5 °C in 5% CO_2_ saturated humidity for 7 days. The cleavage rate was measured 48 h later. A 0.22 μM filter membrane (BIOFIL, Guangzhou, China) was used for filtration. The membrane was pre-warmed in an incubator (Thermo Fisher Scientific, Waltham, MA, USA) at 38.5 °C and 5% CO_2_ for more than 2 h before use. Each experiment was repeated three times.

### 2.3. Immunofluorescence

In order to evaluate the assembly and arrangement of the spindle and chromosomes of oocytes (by software “imageJ 1.53a”), after the removal of cumulus cells by Hyaluronidase, 30 COCs were first fixed at room temperature with 4% PFA (Biosharp, Beijing, China) for 30 min, then permeated with 0.5% TritonX-100 for 30 min, and sealed in DPBS with 1% BSA for 1 h. The oocytes were incubated with α-tubulin (Beyotime, Shanghai, China), a primary antibody, at 4 °C overnight, and the incubated oocytes were incubated with a fluorescent secondary antibody (Beyotime, Shanghai, China) at 4 °C for 2 h. Finally, the oocytes were stained with DAPI (Beyotime, Shanghai, China) for 5 mins. After the nuclear staining, the oocytes were washed with D-PBS 3 times, and then transferred to A laser confocal microscope (Nikon, Shanghai, China) for fluorescence detection. Each experiment was repeated three times.

### 2.4. Fat Drop Dyeing

Thirty sheep oocytes were collected and fixed at room temperature in 4% PFA for 30 min, cleaned with Dulbecco phosphate buffer (DPBS) 3 times, and transferred to BODIPY493/503 (MedChemExpress, South Brunswick, NJ, USA) containing 10 μg/mL at room temperature and incubated for 1 h away from light, and lipid droplets in oocytes were stained. After staining, the oocytes were cleaned with DPBS 3 times and transferred to a confocal laser microscope for fluorescence detection of lipid droplets. Each experiment was repeated 3 times.

### 2.5. ROS and GSH Detection

To assess the ROS and GSH levels in oocytes (by visual method), cumulus cells were removed from 30 COCs using Hyaluronidase and placed in a fluorescent dye containing 500 μL10 μM dichlorofluorescein diacetate (DCFHDA, Thermo Fisher Scientific, Waltham, MA, USA) and 20 μM 4-chloromethyl-6 (Thermo Fisher Scientific, Waltham, MA, USA), 8-difluoro-7-hydroxycoumarin (CMF2HC, Thermo Fisher Scientific, Waltham, MA, USA). The oocytes were incubated in a dark environment at 38.5 °C and 5% CO_2_ saturated humidity for 30 min, washed with DPBS 3 times, and transferred to a fluorescence microscope for fluorescence detection. Each experiment was repeated 3 times.

### 2.6. Real-Time Quantitative RT-PCR

Total RNA was extracted from 50 sheep oocytes using the RNAprepPureMicroKit (Tiangen, Beijing, China). After extraction, the concentration of 1 μL RNA was determined using a spectrophotometer (Nikon, Shanghai, China). Reverse transcription was then performed according to the HiFiScriptcDNASynthesisKit manual for first-strand DNA synthesis. RT-PCR was performed according to the UItraSYBRMixture instructions. The expression richness of each transcript was calculated using the 2^−ΔΔCt^ formula, and β-actin was used as the internal reference gene. The primer sequence is shown in Table 1.

### 2.7. Statistical Analysis

All data were analyzed by single factor ANOVA using GraphPadPrism 8.0 software, and *t*-test was used for comparison between the two groups to generate pictures. *p* > 0.05 means no significant difference, *p* < 0.05 means significant difference, and *p* < 0.01 means very significant difference. Each trial was repeated at least 3 times and the results were expressed as the mean ± standard deviation.

## 3. Results

### 3.1. Effects of Increasing PPARγ Activity on Ovine Oocyte Maturation

Ovine oocytes were treated with 0, 5, 10, and 20 μM RSG, with 0 μM as the control group. As shown in Table 2 and Figure 1, compared with the control group, the maturation rate of ovine oocytes was extremely significantly increased after the addition of a 5 μM RSG treatment (*p* < 0.01), and the proportion of oocytes with normal spindle and mitochondria arrangements was also significantly increased in the 5 μM RSG group (*p* < 0.05). The results indicated that increasing PPARγ activity could improve the maturation rate of ovine oocytes. The optimal drug concentration was determined.

### 3.2. Effects of Increasing PPARγ Activity on Sheep Paracrine Factors

As shown in Figure 2, compared with the control group, the mRNA expression levels of the *GDF9* and *BMP15* genes in the COCs in 5 μM RSG group were significantly increased (*p* < 0.05), indicating that increasing PPARγ activity can effectively improve the quality of IVM ovine oocytes.

### 3.3. Effects of Increasing PPARγ Activity on Oxidative Stress Level of Sheep Oocytes

As can be seen from Figure 3, compared with the control group, the ROS levels in the COCs in the 5 μM RSG group were significantly decreased (*p* < 0.01), the GSH levels were significantly increased (*p* < 0.05), and the mRNA expressions of SOD and CAT were significantly increased (*p* < 0.05). In addition, an increase in the level of cell apoptosis was also detected, and the mRNA expression of BCL2 was significantly increased (*p* < 0.05). The results indicated that increasing PPARγ activity could effectively improve the antioxidant capacity of oocytes and reduce the levels of apoptosis in the ovine oocytes.

### 3.4. Effects of Increasing PPARγ Activity on Lipid Content of Sheep Oocytes

As can be seen from Figure 4, compared with the control group, the content of lipid droplets in COCs of 5 μM RSG group was significantly increased (*p* < 0.05), while the size of the lipid droplets was not significantly different (*p* > 0.05).

### 3.5. Effects of Increasing PPARγ Activity on Lipid Metabolism-Related Genes in Sheep Oocytes

As shown in Figure 5, after increasing PPARγ activity, the mRNA expression of lipid metabolism-related genes changed as follows: the mRNA expressions of *ACAA1* and *PLIN2* were significantly increased (*p* < 0.01), the mRNA expressions of *ACADL* and *CPT1A* were significantly increased (*p* < 0.05), and the mRNA expressions of *ACADM*, *CPT1C*, *CPT2* and *ACSL1* were not changed (*p* > 0.05).

### 3.6. Effects of Increasing PPARγ Activity on Parthenogenetic Activated Embryo Development

As shown in Figure 6, the effect of increasing PPARγ activity on ovine oocyte development to blastocyst was examined. The results showed that after PPARγ increased the level of lipid metabolism of oocytes, the number of developing blastocysts increased significantly (*p* < 0.05), and the mRNA expression of the blastocyst totipotency gene *OCT4* increased significantly (*p* < 0.05).

## 4. Discussion

As an important economic animal in animal husbandry, sheep’s low pregnancy rate is one of the reasons for its low fertility. In vitro embryo production has been widely used to increase fertility in livestock animals, so it is particularly important to provide high-quality oocytes and early embryos through the IVM process. In oocytes, lipids provide energy primarily in the form of β-oxidation. When fatty acids were inhibited from entering the mitochondria for β-oxidation, the development of bovine oocytes and early embryos is impaired due to the lack of sufficient ATP [2]. Fatty acid storage can also improve oocyte quality and enhance embryonic development. For example, during bovine oocyte IVM, supplementation with oleic acid can increase the oocyte lipid content and improve its developmental ability [12]. Melatonin supplementation can improve the lipid metabolism level of porcine oocytes so as to provide sufficient energy for maturation [13]. In addition, the proper assembly and arrangement of chromosomes and spindles during meiosis is essential to avoid chromosome misarrangement and the production of aneuploid gametes [14]. PPARγ, as a lipid sensor, plays an important role in the regulation of lipid metabolism. It has been shown that increasing PPARγ activity in porcine oocytes can increase the maturation rate of oocytes. In this study, the PPARγ activator RSG was used to evaluate the effect of increased PPARγ activity on the IVM of sheep oocytes. The results showed that 5 μM RSG significantly improved the first polar body excretion rate and the proportion of spindle chromosome correct assembly and arrangement, thereby improving the maturation rate of sheep oocytes.

ROS, as a byproduct of mitochondrial β-oxidation, was initially thought to impair cell growth. Studies have shown that normal physiological levels of ROS are essential for cell growth. For example, ROS acts as a second messenger to regulate the activity of macromolecular substances, including metabolism, signaling pathway enzymes, and cytoskeletal networks, in addition to initiating cell responses to environmental signals through ROS signaling. When ROS levels are too high, they can lead to oxidative stress and a series of adverse reactions, resulting in growth arrest and apoptosis [15]. Oocytes regulate REDOX stability during maturation and early embryonic development through ROS. The ROS levels in cells are regulated by superoxide dismutase (SOD), catalase (CAT), and glutathione reductase (GPX). The free O^2−^ is converted to H_2_O_2_ by SOD, and the H_2_O_2_ is decomposed into H_2_O and O_2_ by CAT and GPX. Ren et al. found in rat cardiomyocytes that after overexpression of PPARγ, *BCL2* was upregulated, ROS levels were reduced, and thus the apoptosis of cardiomyocytes was inhibited under oxidative stress [16]. The overexpression of PPARγ upregulated the expression of *BCL2* in neurons, thereby protecting cells from apoptosis caused by oxidative stress [17]. Therefore, PPARγ plays an important role in maintaining the equilibrium of cellular oxidative homeostasis. In this study, increasing PPARγ activity in ovine oocytes can effectively reduce ROS levels in oocytes and reduce cell apoptosis.

During maturation, oocytes release a soluble growth factor, oocyte secretion factor (OSFs), that regulates the function of surrounding cumulus and granulosa cells. The growth differentiation factor 9 (GDF9) and bone morphogenetic protein 15 (BMP15), which belong to the growth transforming growth factor β (TGFβ) superfamily, have been the most extensively studied. GDF9 is essential for follicle formation, and the formation of follicles in mice and sheep with GDF9 deficiency is hindered [18,19]. Cumulus cells wrap around oocytes, thus realizing a “dialogue” between the cumulus and ova. Cumulus cells participate in oocyte maturation and improve its developmental potential by transferring small molecules to ova. In turn, oocytes regulate the proliferation and expansion of cumulus cells through paracrine factors. *GDF9*, as a paracrine factor, is involved in regulating the expansion of cumulus cells around sheep oocytes [20]. A lack of BMP15 in mice did not affect ovulation [21], whereas sheep lacking BMP15 did not ovulate normally [22], which may be related to differences between species. During oocyte development, GDF9 and BMP15 synergistically regulate oocyte development, which is of great significance for maintaining normal fertility.

Lipid metabolism plays an important role in the oocyte IVM process. There is considerable interspecific variation in lipid content in mammalian oocytes; for example, pig, bovine, and sheep oocytes appear darker because they contain more lipids, while human and mouse oocytes appear more transparent because they contain less lipids. The accumulation of lipids was also observed during the maturation of ovine oocytes [23]. In order to maintain energy homeostasis in cells, fatty acids produced by LDs through lipolysis are absorbed into mitochondria for β-oxidation [24]. In this process, a close proximity between LDs and mitochondria is clearly observed [25]. This further illustrates the importance of lipids as energy substrates during oocyte maturation. In this study, after increasing PPARγ activity, the number of LDs in sheep oocytes increased, the volume did not change, and the maturation rate of oocytes was improved. This may indicate that a moderate increase in lipid content is beneficial to oocyte maturation.

The maturation process of mammalian oocytes is often accompanied by a large amount of energy consumption, so the level of metabolism in the cell is more vigorous. The energy produced by glucose metabolism is insufficient to maintain the needs of oocytes during maturation, at which time fatty acids are required to enter the mitochondria for β-oxidation to produce more energy to meet this demand. Next, we studied the mRNA expression of genes involved in fatty acid oxidation after increasing PPARγ activity. Acetyl-coa acyltransferase 1 (ACAA1) is involved in the extension and degradation of fatty acids by catalyzing the last step of the β-oxidation pathway and is an important regulatory gene in the process of cellular lipid metabolism. When the expression of ACAA1 was inhibited, the β-oxidation level in sheep adipocytes was decreased [26], while the β-oxidation of fatty acids in mouse liver was enhanced after the expression of *ACAA1* gene was up-regulated. Long-chain acyl-CoA dehydrogenase (ACADL) and medium-chain Acyl-CoA dehydrogenase (ACADM) belong to the Acyl-CoA dehydrogenase family, where ACADL is A mitochondrial enzyme that catalyzes the initial steps of fatty acid oxidation. The knockout of *ACADL* in mice showed frequent pregnancy loss and was accompanied by lipid deposition in the livers and hearts of born mice [27]. In addition, ACAA1 has been shown to be involved in the regulation of lipid metabolism in sheep cells [28]. Carnitine palmitate transferase (CPT) catalyzes long-chain fatty acids from cytoplasm into mitochondria for fatty acid oxidation and is an important fatty acid rate-limiting enzyme, including CPT1 acting outside the mitochondrial membrane and CPT2 acting inside the mitochondrial membrane. The protein in LDs is mainly composed of peritlipoprotein (PLIN1-5) encoded by five different genes, in which the expression level of PLIN2 is correlated with TG content and lipid droplet density [29]. The expression of PLIN2 increased during oocyte maturation and early embryo development [30]. In this study, the mRNA expression levels of *ACAA1*, *ACADL*, *CPT1A,* and *PLIN2* were increased after increasing PPARγ activity. Combined with the increase in the above lipid content, it was proved that the increase in PPARγ activity improved the lipid metabolism of sheep oocytes and enhanced the mitochondrial β-oxidation level.

*OCT4*, *SOX2,* and *NANOG* are involved in maintaining pluripotency and the self-renewal of embryonic cells. A number of studies have shown that *OCT4* works with *SOX2* and *NANOG* to form and maintain the pluripotency and self-renewal of embryonic stem cells [31,32,33,34]. Rodda et al. believed that *SOX2-OCT4* belonged to the top level of pluripotent gene regulation, and they jointly controlled *NANOG* to regulate cellular pluripotency through downstream genes [35]. When *OCT4* was knocked out, early embryonic development failed in the presence of *NANOG* [36]. In addition, *OCT4* is essential for maintaining the pluripotency of the inner cell mass [37,38]. In this study, the number of parthenogenetically activated ovine oocytes developing into blastocysts increased and, after increasing PPARγ activity, the number of parthenogenetically activated sheep oocytes that developed into blastocysts on the seventh day increased, and the mRNA expression of *OCT4* also increased.

## 5. Conclusions

In conclusion, increasing PPARγ activity during ovine oocyte IVM has positive effects on oocyte maturation and parthenogenetic embryo development. The detection of lipid droplet staining and lipid metabolismrelated genes showed that increasing PPARγ activity could improve lipid metabolism levels, provide sufficient energy for oocyte maturation, and have positive effects on subsequent embryonic development.

## Figures and Tables

**Figure 1 vetsci-11-00397-f001:**
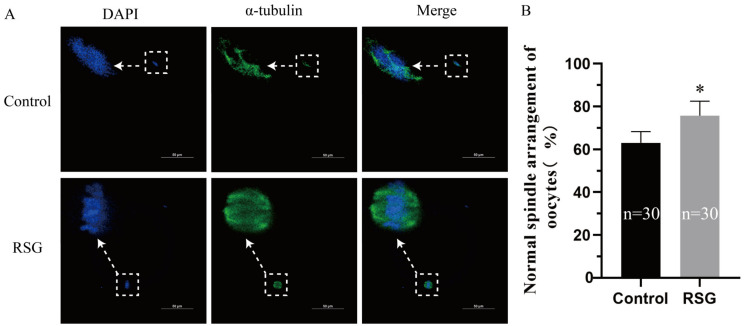
Effects of RSG on oocyte (MII stage) maturation. (**A**) Representative images of the spindles of oocytes treated with RSG by immunofluorescence staining. (**B**) Ratio of normal spindles in RSG-treated oocytes. “*” indicates significant difference.

**Figure 2 vetsci-11-00397-f002:**
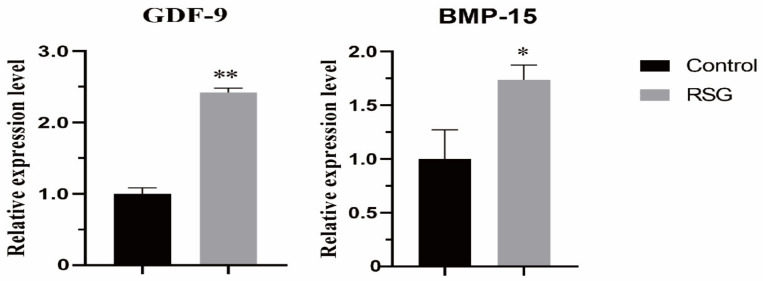
Effect of RSG treatment on mRNA expression of paracrine-related genes in oocytes. “*” indicates significant difference; “**” indicates difference is extremely significant.

**Figure 3 vetsci-11-00397-f003:**
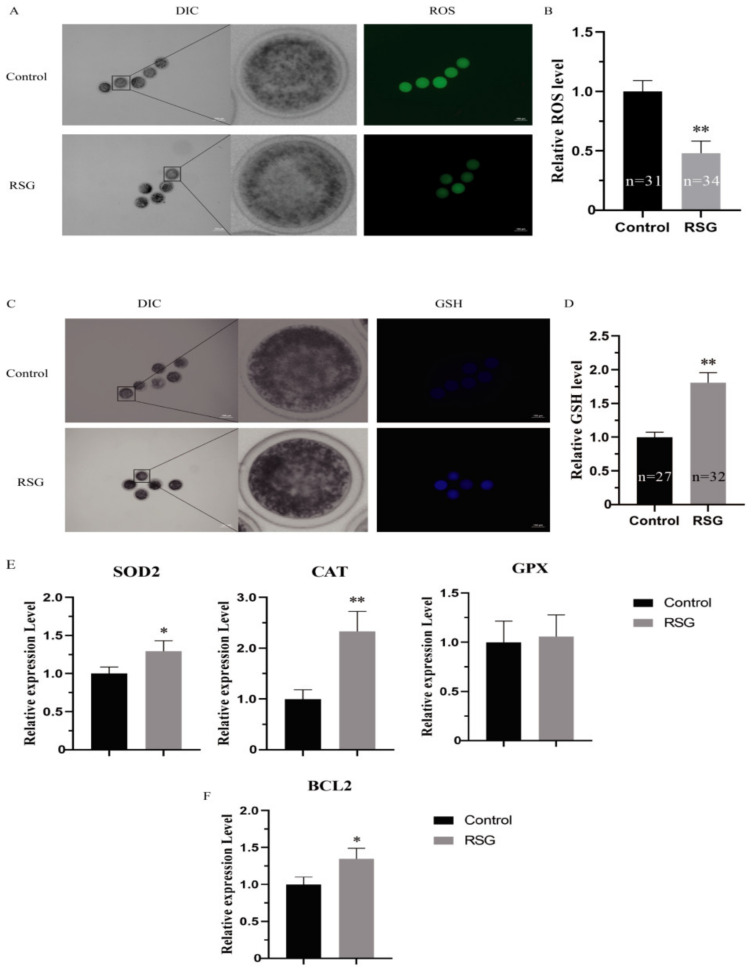
RSG can effectively improve the antioxidant capacity of oocytes and reduce the apoptosis level of ovine oocytes. (**A**) Representative images of ROS staining of oocytes treated with RSG. (**B**) Relative fluorescence intensity of ROS in RSG-treated oocytes. (**C**) Representative images of GSH staining in RSG-treated oocytes. (**D**) Relative fluorescence intensity of GSH in RSG-treated oocytes. (**E**) Expression levels of oxidative stress-related genes in RSG-treated oocytes. (**F**) Expression levels of anti-apoptotic genes in RSG-treated oocytes. “*” indicates significant difference; “**” indicates difference is extremely significant.

**Figure 4 vetsci-11-00397-f004:**
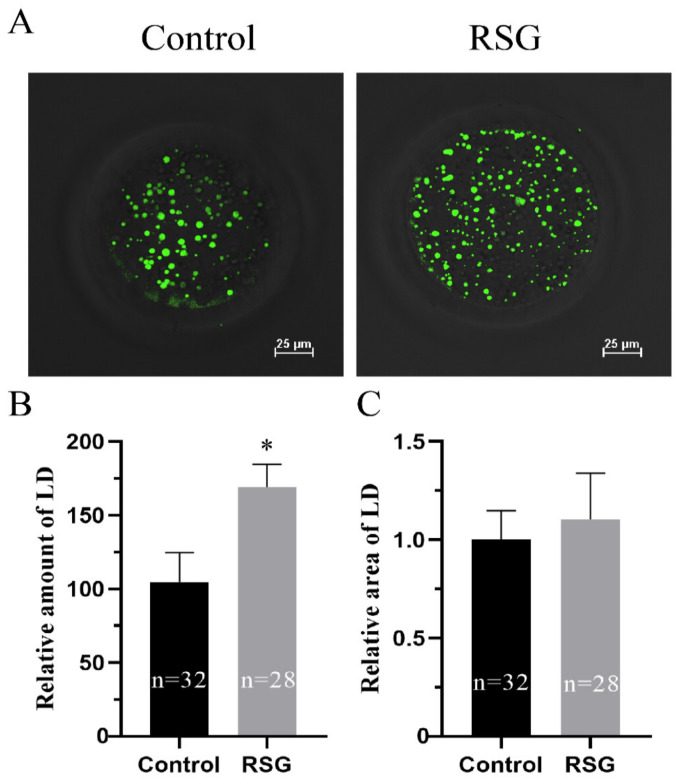
Effects of RSG treatment on lipid content of oocytes. (**A**) Effects of RSG on lipid droplet staining. (**B**) Effects of RSG on the number of lipid droplets. (**C**) Effects of RSG on the area of lipid droplets. “*” indicates significant difference.

**Figure 5 vetsci-11-00397-f005:**
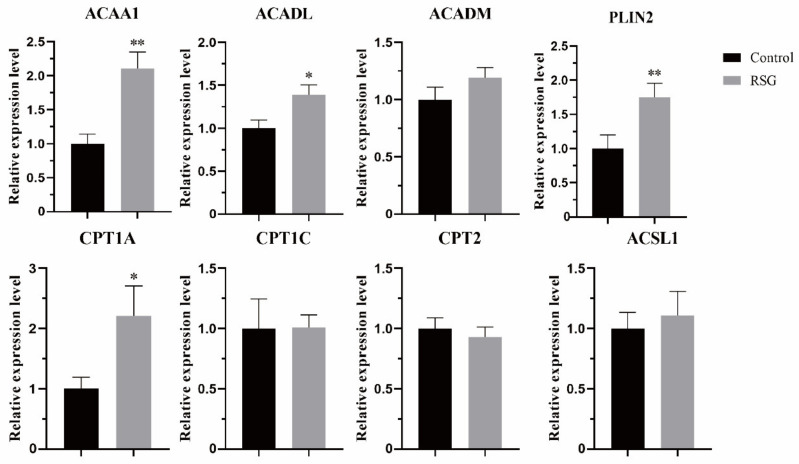
Effect of RSG treatment on mRNA of genes related to lipid metabolism in oocytes. “*” indicates significant difference; “**” indicates difference is extremely significant.

**Figure 6 vetsci-11-00397-f006:**
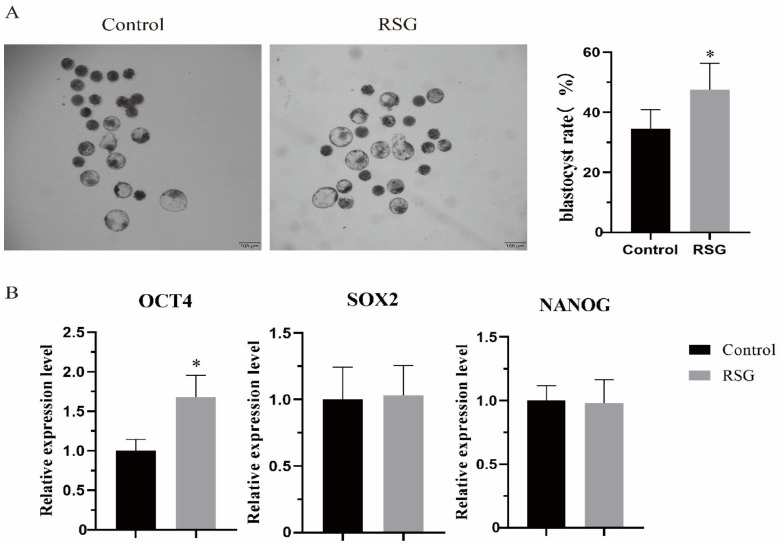
Effects of RSG treatment on parthenogenetic activated embryo development. (**A**) Representative picture of RSG-treated blastocyst development and normal blastocyst rate. (**B**) RSG-treated mRNA of genes involved in maintaining embryonic cell pluripotency and self-renewal. “*” indicates significant difference.

**Table 1 vetsci-11-00397-t001:** Primer sequence information of related genes.

Gene	Secquence	Amplicon Size	GenBank Accession Number
β-actin	F: TCTCTTCCAGCCTTCCTTCCTG R: AGCACCGTGTTGGCGTAGA	184 bp	NM_001009784.3
GDF9 BMP15	F: GACGCCACCTCTACAACACT R: TCCACAACAGTAACACGATCCA F: GCCCAACCAATCACTTTCCTTC R: GCCACCAGAACTCAAGAACCT	133 bp 192 bp	NC_056058.1 NM_001114767.2
OCT4	F: CGTGGTCAGAGTGTGGTTCT R: TGGCTTCAGAGGAAAGGATACG	133 bp	NC_003074.8
SOX2	F: ACATGAACGGCTGGAGCAA R: GCGAGCTGGTCATAGAGTTGTA	156 bp	NC_056054.1
NANOG	F: ACTGTCTCTCCTCTTCCTTCCT R: CTCTTCCTTCTCTGTGCTCTCC	120 bp	NC_056056.1
CPT2	F: GCCACCTATGAGTCCTGTAGC R: CATCGCTGCTTCTCTGGTCA	195 bp	NC_056054.1
ACSL1	F: TCTGGATAAGGACGGCTGGTT R: AGGTTCACTTCGCTGGTAGATG	160 bp	NC_056079.1
ACSL3 ACAA1	F: ACCATCGCCATCTTCTGTGAA R: GGACCGCCTAGAGTAGCATAC F: AGGAGGTCCAAGGCAGAAGAG R: TCCACATCGTCCACCGTCAG	107 bp 155 bp	NC_056055.1 NC_056072.1
ACADM ACADL	F: GTGGAGGTCTTGGACTTGGAA R: ACACAATGGCTCCTCAGTCAT F: TGTGACACTGTGATCGTCGTAG R: CTGCTGGCAACCGTACATCT	182 bp 184 bp	NC_056054.1 NC_056055.1
CPT1A	F: TCACATCCAGGCGGCAAGA R: GAGCAGAGCGGAATCGTAGAC	120 bp	NC_056074.1
CPT1B	F: TGCGTTCTTCGTGGCTCTG R: GCGTGCTCTGTGTTGAGTC	177 bp	NC_056056.1
CPT1C	F: AATCCACCATGACTCGCTTGT R: CCACTCATCGCTGCCTTCA	181 bp	NC_056067.1

**Table 2 vetsci-11-00397-t002:** In vitro culture maturation rate of ovine oocytes in each group.

Group	Number of Oocytes	PB Rate(%)
0	121	43.52 ± 5.03 ^b^
5 μM	123	54.29 ± 5.13 ^a^
10 μM	106	43.86 ± 4.51 ^b^
20 μM	99	40.33 ± 5.57 ^b^

The bulleted values represent the mean ± SD. The different letters indicate statistically significant differences (*p* < 0.05).

## Data Availability

The datasets presented in this study can be found in the article.

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
