# Peer review of "Effects of Exogenous Regulation of PPARγ on Ovine Oocyte Maturation and Embryonic Development In Vitro"

_vetsci, 2024, doi:10.3390/vetsci11090397_

Round 1

Reviewer 1 Report

Comments and Suggestions for Authors

1. Please add the primer information for β‐actin as the internal reference gene in Table 1, as it is mentioned in line 137 of the manuscript but not included in the table.

2. Clarify the statement in line 143: “All data wer2.7. Statistical Analysis”.

3. In Figure 1, it is shown that the control group has about 60 per cent of normal spindle. What stage was the oocyte selected for this experiment? Also, please explain what should be the minimum percentage of normal spindle for normal development of oocytes? Will the oocytes selected for this experiment affect the effectiveness of their subsequent IVM compared to normally developing oocytes?

4. The significant increase in the relative expression of the GDF9 and BMP15 genes described in Section 3.2 suggests an effective improvement in the quality of IVM in sheep oocytes. Can the relative expression of these genes alone account for this improvement? Are there other indicators that could help to illustrate this?

5. Explain the discrepancy in line 177: “the mRNA expressions of SOD and GPX were significantly increased (P < 0.05),” whereas Figure 3E shows no significant increase for GPX.

6. Clarify the difference between the description of PPAR activity in sections 3.2 and 3.3 and the PPARγ activity described elsewhere in the manuscript.

7. Confirm whether genes names and P in the manuscript need to be italicized and make necessary changes throughout the text.

8. Confirm whether the blastocyst rate statistics in Figure 6A are from different experimental batches and provide the total number of oocytes counted.

9. In the file vetsci-3109681-original-images, the comments in Figure 3 and Figure 4 are identical. Please revise them.

Comments on the Quality of English Language

Please pay attention to grammar errors in the text. 

Author Response

Please see the attachment." in the box if you only upload an attachment.

Reviewer 2 Report

Comments and Suggestions for Authors

The paper presents interesting data, but before considering it for publication, it requires some changes. First of all, the general English language is suitable and easy to understand, but the professional terms are often missing or used inappropriately. The authors need to check other published papers and correct this. Furthermore, both the Introduction and Discussion need some work. The information provided is okay, but the content is quite disorganized. The authors mention the same things several times without fully addressing the topic, e.g., oocyte maturation, melatonin, lipid metabolism, energy required by oocytes during maturation, etc.

Specific comments:

Please change:

Line 41: raw foam rupture

Line 42: discharge (extrusion?)

Line 44: what means qualitative? (cytoplasmic?)

Line 45-47: please provide citation

Line 47: ‘their’ what does it stands for?

Line 49: please change implantation to development

Line 53: what authors mean by ‘quality control’

Line 57 and 45: there are contradicting statements: weak glycolysis and mainly glycogen – please check and improve

M&M

Please provide information on where all chemicals and media were purchased. Additionally, include the media content in brackets immediately after they are mentioned (see below). Please add the number of oocytes investigated in each experiment and the number of replicates performed. Please check if DPBS is Dulbecco….? or Duchennean…?

Line 85: what is egg collection fluid?

Line 86: change: discharged and the same Line: what provided the 37 C dgr temperature?

Line 90: move the information on the culture media content to the line when you mention it for the first time. Same for SOF in next paragraph

95: how maturation was recognised?

116 and 130: how it was detected? Must visually or you used software?

All the tables, figures have to be mentioned in the text of the results where appropriate

Table 2 – explain what is PBE

Discussion:

229: melatonin – provide citation

273: what ‘it’ stands for?

293-297: unnecessary here?

Comments on the Quality of English Language

As mentioned above: the general English language is suitable and easy to understand, but the professional terms are often missing or used inappropriately.

Reviewer 3 Report

Comments and Suggestions for Authors

Simple receptor activators can significantly improve the quality of oocytes maturation and the development of parthenogenetic embryos. This article provides a good research report, which is worth our reading and reference. However, further thinking is how this article can help us improve the quality of oocyte culture in vitro, such as improving the effective composition of the culture medium, etc.! Such improvements could be fundamental. If possible,It is recommended to add a comparison chart of blastocyst cell counts for group tests

Author Response

Thanks for your suggestion, we will consider further analysis of the composition of the culture solution and the number of blastocyst cells in subsequent experiments.

Round 2

Reviewer 2 Report

Comments and Suggestions for Authors

Authors answered to all my comments